# Microstructure and Mechanical Properties of Ti/Al–SiC/Ti Clad Plates Prepared via the Powder-in-Tube Method

**DOI:** 10.3390/ma16175986

**Published:** 2023-08-31

**Authors:** Xianlei Hu, Qincheng Xie, Yi Yuan, Ying Zhi, Xianghua Liu

**Affiliations:** 1State Key Laboratory of Rolling and Automation, Northeastern University, Shenyang 110167, China; hu_xianlei@263.net (X.H.); yuanyi17302415326@163.com (Y.Y.); zhiying@ral.neu.edu.cn (Y.Z.); 2Suzhou Dongbaohaixing Metal Material Science and Technology Co., Ltd., Suzhou 215625, China; 3School of Material Science and Engineering, Northeastern University, Shenyang 110167, China

**Keywords:** SiC particles, powder-in-tube method, Ti/Al–SiC/Ti clad plate, mechanical property

## Abstract

SiC particle-reinforced Ti/Al/Ti clad plates were successfully fabricated by the powder-in-tube method. The surface micrography, element diffusion, peeling strength and tensile property of clad plates were studied after annealing and cold rolling. The experimental results show that 6 wt.% is optimal. The presence of SiC particles has been seen to significantly enhance the diffusion of Ti and Al elements. Additionally, it has been observed that the diffusion width of the intermetallic compound (IMC) increases as the size of SiC particles grows. However, it is worth noting that the average of Ti/Al–SiC/Ti clad plates initially increases and subsequently falls. The optimized diffusion thickness of the Ti/Al–SiC/Ti clad plate’s IMC layer determined via the powder-in-tube method is approximately 4.5 μm. The 1 μm SiC-reinforced Ti/Al/Ti clad plate can obtain the best mechanical properties after annealing at 500 °C and further hot rolling, and the peeling strength, ultimate tensile strength and elongation are 31.5 N/mm, 305 MPa and 26%, respectively. The efficacy of Ti/Al–SiC/Ti clad plates in delivering exceptional performance is substantiated by the analysis of peeling surfaces, peeling tests and tensile testing, which collectively demonstrate the presence of compact and homogenous intermetallic compounds.

## 1. Introduction

Currently, numerous burgeoning industries are experiencing rapid growth, necessitating the development of alternative materials as traditional single-metal materials fail to adequately fulfil the demands of national production and daily life. The complicated service environment requires that metal structural materials have composite properties such as light weight, corrosion resistance, robustness and high temperature resistance [1,2,3]. For this reason, composites of different metals are preferred. Aluminum has low density, excellent electrical and thermal conductivity and corrosion resistance and meets the requirements of many commercial uses, but the strength and stiffness of aluminum and its alloys are low. Titanium and its alloys have the advantages of high specific strength, high temperature resistance and good fracture toughness, becoming the ideal material for industrial applications, but titanium alloys are very expensive [4,5].

Clad plates composed of titanium (Ti) and aluminum (Al) exhibit exceptional material qualities inherent to both metals. Consequently, the stamped components derived from such plates hold significant potential for diverse applications across industries including automotive, marine, aircraft, electronics, medical and other industries. Based on the Ti–Al binary phase diagram, it is seen that the solid solubility of aluminum (Al) in titanium (Ti) is higher compared to the solubility of titanium in aluminum. Consequently, when subjected to elevated temperatures, diffusion between the two elements can occur, leading to the formation of intermetallic compounds that facilitate metallurgical bonding.

Ti/Al composites have been successfully fabricated by various methods such as rolling, hot pressing and explosive welding, as acknowledged by researchers both domestically and internationally. During the process of plastic deformation, it is observed that a significant residual stress is present at the interface of the composite material. In order to enhance the mechanical characteristics and enable future forming, it is common practice to modify the structure of the interface using annealing treatment [6]. According to the findings of Jafari et al. [7], annealing demonstrates a favorable influence on the control of bonding strength at the interface between Ti and Al. In a study conducted by Assari. AH et al., an investigation was carried out on a composite sheet consisting of hot-press Ti/Al. The researchers observed that the sheet predominantly consisted of coarse crystal Al, fine crystal Ti and TiAl_3_ phases. Furthermore, it was noted that the composite sheet exhibited remarkable yield strength and elongation properties [8]. I.A. Bataev et al. analyzed the growth characteristics of the intermetallic compound layer after the annealing of a Ti/Al composite plate constructed by explosive welding, found that TiAl_3_ was the only intermetallic compound and studied the effect of TiAl_3_ on the impact strength of the composite plate [9]. Although the preparation technology of Ti/Al/Ti clad plate has been extensively explored, the above preparation technology still has defects. For example, Ti/Al composites prepared by the rolling composite process have low bond strength and poor performance stability. Composite plates produced by the explosive welding method have high interface strength, but the production efficiency is limited, and there is a certain risk. Cladding welding technology easily causes environmental pollution and other problems.

In contrast to conventional preparation techniques, the PIT method offers several advantages, including streamlined equipment and enhanced production efficiency. In their study, H. Kumakura et al. were able to successfully produce MgB2 superconducting strips using the powder-in-tube method. Similarly, Chiba. R et al. successfully fabricated 1 mm thick aluminum alloy strips using powder sleeve technology, and the strength and density were greatly improved compared to unprocessed aluminum alloy [10,11]. In a study conducted by H.T. Gao, the PIT approach was employed to effectively fabricate Cu/Al/Cu cladding bands. The results indicated that Cu/Al/Cu cladding bands with CuAl_2_ diffusion layers exhibiting high density and uniformity demonstrated exceptional performance [12].

The utilization of the powder-in-tube (PIT) method for fabricating composite plates with an aluminum powder core material offers significant design flexibility for aluminum powder components. H. Springer investigated the intermetallic compounds present in diffusion-treated groups of carbon steel/pure aluminum and carbon steel/aluminum–silicon alloy. The research employed various methodologies, including optical microscopy (OM), scanning electron microscopy (SEM), electron backscatter diffraction (EBSD), transmission electron microscopy (TEM) and additional techniques as referenced in source [13]. At a temperature of 675 °C, the inhibition of intermetallic compound formation at the interface of the solid–liquid diffusion pair can be achieved with an aluminum alloy containing 5 wt.% of silicon. Nevertheless, when subjected to a solid–solid diffusion pair at a temperature of 600 °C, the rate of formation of intermetallic compounds in Al–Si alloys with a Si content of 5% exhibited an acceleration. The inhibition mechanism of intermetallic compounds at the interface of rolled steel–aluminum composites was explored by J.L. Cao et al. It was found that the inclusion of silicon (Si) effectively hinders the mutual diffusion of iron–aluminum (Fe–Al) [14]. At a temperature of 580 °C, the presence of silicon in alloys at a mass fraction of 6.0% leads to an increased impact of inhibition. Silicon carbide-reinforced aluminum matrix composites have gained significant traction across various disciplines due to their notable attributes, including corrosion resistance, high strength and metallization. Mondal. DP et al. found that a high SiC_p_ concentration (approximately 15%) reduces the elongation and fracture stress of the composite [15]. V. Erturun. et al. [16] conducted an investigation on the influence of SiC particle size on the properties of composites. They devised a powder metallurgy technique to fabricate SiC/Al composites.

According to our literature review, the effect of SiC concentration on the formation of intermetallic compounds at the Ti–Al interface during the preparation of Ti/Al–SiC/Ti composite plates has been the subject of few investigations. The effect of SiC particle size on the growth of the intermetallic layer at a given concentration must also be investigated. The effects of SiC content and particle size on the microstructure properties of the hot-rolled composite Al–Si powder/TA1 titanium tube interface was studied by diffusion annealing treatment under the same conditions, and the mechanism of action of SiC particles was discussed in greater detail.

To the best of our knowledge, the open literature suggests that no attempt has been made to study Ti/Al–SiC/Ti clad plate produced by the powder-in-tube method. To fill this gap, the interface and mechanical strength of Ti/Al–SiC/Ti clad plates fabricated by this novel method under different reduction rates and SiC particles were investigated in this paper.

## 2. Materials and Methods

The cladding material utilized in this study consisted of an industrial pipe made of titanium (TAl) with a purity level of 99.5%. The pipe had an exterior diameter of 10 mm and a wall thickness of 1 mm. The core material was atomized aluminum powder and SiC particles, among which SiC particles were used as interface bond-strengthening materials. The particle morphology observed by scanning electron microscopy (SEM) of the atomized aluminum powder is depicted in Figure 1. The Ti tube was heated to 650 °C and held for 20 min to soften it. The inner wall of the titanium tube was cleaned with a wire brush and acetone to remove oxides and other impurities. This method can expose bare metals, roughen the surface and promote metal bonding. Figure 2 shows the preparation process of the Ti/Al–SiC/Al composite plate. In a ball mill, SiC powder of 0.5/1/3 μm particles and 99.99% pure aluminum powder of 10 µm diameter were fully homogenized and named HA500, HS_0.5_, HS_1_ and HS_3_, respectively, and then the homogenized mixed powder was placed into the titanium tube, sealed at both ends, heated to 500 °C and held for 20 min. Tubes with different particle sizes of Al–SiC powder after annealing were thinned to 4 mm thickness on a two-roll reversible flat roller mill, annealed at 500 °C for 5 min and then further rolled into 1.7 mm clad plates. Finally, Ti/Al–SiC/Ti clad plates with smooth surfaces and no cracking at the boundary were prepared by annealing Al–SiC 1.7 mm clad plates with different particle sizes at 550 °C for 60 min under vacuum.

The analysis of the morphology of the bonding interface, peeling surface and tensile fracture was conducted using a SHIMADZU EPMA-8050G electron probe micro-analyzer (EPMA) equipped with an energy-dispersive spectrometer (EDS). In order to examine the diffusion of elements at the interface, an electron probe micro-analyzer (EPMA) model JEOL JXA-8530F (Tokyo, Japan) was employed. The EPMA operated at an acceleration voltage of 20 kV and a sample current of 2 × 10^−8^ amperes. In order to evaluate the adhesive strength and tensile characteristics, a T-type peeling test and a tensile test were conducted using a TH5000 universal testing machine (Tianhui Machinery Co., Ltd., Yangzhou, China). The tests were performed at room temperature with a crosshead speed of 3 mm/min. The standard reference for tensile tests at room temperature is GB/T228.1-2021 [17]. The tensile rate was the same as the peel test rate. Figure 3 depicts images illustrating the peeling test and tensile test sample.

## 3. Results and Discussion

### 3.1. Element Diffusion of Ti/Al–SiC/Ti Clad Plates

As shown in Figure 4, SEM images of Al–SiC powder casing with different particle sizes after annealing of the sealed pipe show that SiC particles are unevenly dispersed in the Al matrix, and obvious SiC agglomeration can be detected in the Al matrix, corresponding to the red arrow mark. In Figure 4a, the agglomeration of SiC particles at 0.5 µm was more significant, and the agglomeration decreased significantly with increasing SiC particle size (Figure 4b,c). When the SiC particles are small, the specific surface area is large, the surface activity is high, the fluidity is good and the agglomeration is easily affected by van der Waals forces. The agglomeration of SiC in the matrix is influenced by the ratio of matrix to reinforced particle size, and a decrease in this ratio has been found to mitigate the incidence of agglomeration [18]. Consequently, the introduction of a higher concentration of SiC particles has been observed to have a notable effect on mitigating the agglomeration of SiC within the matrix.

Figure 5 shows the EPMA interface morphology and element distribution of the Ti/Al–SiC/Ti composite plate. Figure 5b,d,f,h indicates that the thickness of the interfacial diffusion layer of the composite plate gradually decreases with increasing SiC particle size, but the Ti/Al–SiC interface morphology becomes increasingly uniform and continuous. Nano-SiC particles easily form agglomerations. However, compared with large SiC particles, nanoscale particles have a relatively weak hindering effect on the element diffusion of Ti/Al. H.T. Gao studied the influence of SiC particle size on the morphology of the Cu/Al interface and observed similar phenomena [12]. S. Yan et al. found that the reduction of SiC particles, coupled with higher thermal residual stress and larger surface area, led to a decrease in the thermal expansion coefficient of the intermetallic compound layer and the thermal conductivity of Ti/Al [19]. As a result, the intermetallic compound layer became narrower and more uniform. In addition, there are some small cracks in IMCs because, during the rolling process, the TiAl_3_ intermetallic compound layer with poor toughness undergoes brittle fracturing under a high compression rate, and excessive local stress concentration also leads to cracks. There is an obvious stress concentration at the interface of Ti and Al–SiC. This phenomenon can be attributed to the thermal expansion coefficient (SiC ≈ 3.8 × 10^−6^, Ti ≈ 10.03~9.41 × 10^−6^, Al ≈ 23.1 × 10^−6^) and elastic modulus (SiC ≈ 475 GPa, Ti ≈ 106.4 GPa) of SiC particles, significantly different from that of Ti and Al substrates. After cooling to room temperature following hot rolling, this difference results in a significant concentration of stress at the interface of the composite.

### 3.2. Peeling Strength of Ti/Al–SiC/Ti Clad Plates

Figure 6 shows the Ti/Al–SiC/Ti peel strength curve and the average peel strength. Figure 6a shows that the initial peeling strength of the composite plate with SiC particles added is higher than that of the composite plate without SiC reinforcement, indicating that SiC has an excellent interface-strengthening effect. Among the SiC-reinforced composite plates, HS_0.5_ has the lowest initial peel strength due to microcracks in the intermetallic compound layer that worsen the bonding strength of the interface. As shown in Figure 5c, the thicker intermetallic compound layer delays crack propagation, and the peel strength increases significantly with increasing peel distance. The stripping force exhibits a linear reduction within the stripping distance range of 0 to 20 mm, indicating a favorable bonding force at the interface of the Ti/Al–SiC composite. As the stripping distance is extended, there is a noticeable trend of gradual variation in the stripping force. This suggests that the intermetallic compound layers present at the Ti/Al interface of the HS_3_ sample exhibit a continuous bonding behavior. HS_3_ has an initial peeling strength similar to that of HS_1_, indicating that the thin intermetallic compound layer (Figure 5g) can better resist initial tearing. The HS_1_ sample has the highest initial peeling strength; the peeling distance increases, the peeling strength decreases slightly and some parts of the peeling curve fluctuate, which may be due to local microcrack expansion, but the overall curve changes within a stable range, indicating that an appropriate thickness of the intermetallic compound layer can obtain excellent interfacial bonding strength.

Figure 6b shows the average peel strength of the Ti/Al/Ti and Ti/Al–SiC/Ti composite plates. The decrease in the peel strength of HS_0.5_ can be attributed to the smaller SiC particles, which have a poor effect on the diffusion of Ti/Al elements, resulting in the formation of a thicker intermetallic compound layer. In addition, the presence of many microcracks in the intermetallic compound layer adversely affects the bonding properties of the interface. The decrease in HS_3_ interface bonding strength is related to the increase in SiC particle size, which has a strong hindering effect on element diffusion, resulting in an unsatisfactory metallurgical bonding effect. Further analysis of the fracture interface morphology can provide insight into the reasons for the change in interface bonding strength.

The effect of the interface state on the properties of composites is significant. The X-ray diffraction (XRD) technique was employed to analyze the titanium and aluminum surfaces subsequent to the peeling strength test, as depicted in Figure 7. The main components are α-Al, Ti, TiAl_3_ and SiC. For the peeling surface of the Ti side, the contents of TiAl_3_, Ti, Al and SiC decreased with the rise in SiC size, whereas for the Al side, the contents of Ti and Al showed the same change trend, and the content of TiAl_3_ and SiC did not change significantly with the increase in SiC size. Therefore, an increase in the size of SiC particles minimizes the generation of intermetallic compounds. In addition, the SiO_2_ oxide layer present on the surface of SiC particles may react with Al:(1)$$3{\mathrm{SiO}}_{2}+4\mathrm{Al}\to 2{\mathrm{Al}}_{2}O_{3}+3\mathrm{Si}$$

In XRD analysis, in addition to the Ti and Al matrix, only TiAl_3_ and SiC were detected because the SiC particles and Al powder were pretreated with 5% HF pickling before mixing. The quantitative analytical results of X-ray diffraction (XRD) on the titanium (Ti) and aluminum (Al) sides of the Ti/Al–SiC/Ti composite are presented in Table 1.

### 3.3. Morphology of the Bonding Interface after Peeling

Figure 8 shows the peel interface morphology of the Ti side of Ti/Al/Ti clad plates strengthened with SiC particles of different sizes. In Figure 8a,c,e, it can be observed that the inner surface of the Ti side is bonded to pure Al, which is due to the good consolidation quality of Ti and Al and the high bonding strength between Ti, Al and SiC, resulting in the direct tearing of the inner part of the Al matrix during the stripping test, and part of the pure aluminum is bonded to the Ti matrix. The microstructure also shows that there are dimples on the surface of the Ti matrix (Figure 8b,d,f).

Figure 8a shows the low-magnification morphology of the Ti matrix peel surface of the HS_0.5_ sample. It can be clearly observed that pure Al is not evenly distributed on the surface of the Ti matrix, and some parts of the surface are smooth without Al adhesion. This is because the bonding strength of Ti and Al is less than the strength of pure Al itself, and the Al side is completely separated from the Ti side under the action of the stripping force. As shown in Figure 8b, many large and shallow dimples existed on the surface of the Ti matrix under high multifold observation, and scattered pieces of TiAl_3_ were present in the dimples. This was because during the stripping test, the peeling force caused the thick intermetallic compounds to break up, and during the tearing process, they were randomly scattered in the dimples on the surface of the Ti matrix.

Figure 8c,e describes the tear surface morphology of Ti/Al/Ti composites strengthened with micro-SiC. Compared with HS_0.5_, the number of dimples on the surface of the Ti matrix of the HS_1_ and HS_3_ samples is significantly increased, and there is no agglomeration or TiAl_3_ phase fracturing at the interface. At low magnification, the tear morphology of the HS_3_ and HS_1_ samples showed high similarity. Figure 8f shows the Ti matrix tear surface morphology of the HS_3_ sample at high magnification. Significant amounts of lumpy TiAl_3_ phase and SiC aggregation were observed in the dimple. These particles obviously hinder the mutual diffusion of Ti/Al at the interface, forming a thin intermetallic compound layer, resulting in reduced thermal conduction and a metallurgical bonding effect and weakening the interfacial bonding strength of the Ti/Al matrix. Table 2 presents the results of the SEM–EDS microchemical composition study conducted on the HA_0.5_, HA_1_ and HA_3_ specimens.

### 3.4. Tensile Strength of Ti/Al–SiC/Ti

Table 3 and Figure 9 show the tensile properties of different sizes of SiC-reinforced Ti/Al/Ti composite plates. The ultimate tensile strengths of the 0.5 μm, 1 μm and 3 μm composite sheets were 270 MPa, 305 MPa and 296 MPa; the yield strengths were 195 MPa, 220 MPa and 211 MPa; and the elongations were 20.7%, 26% and 27.1%, respectively. The results show that the addition of SiC particles can significantly improve the tensile strength, yield strength and elongation. The 1 μm SiC-reinforced Ti/Al/Ti clad plate had maximum strength and elongation. The chemical composition and thickness of the intermetallic compound layer significantly affect the mechanical properties of Ti/Al–SiC/Ti. SiC particles (6 wt.%) promote the formation of intermetallic compounds and reduce the Ti/Al interface bonding strength, but the strength of the composite plate is improved by increasing the strength of the Al matrix. The good composition of Al and SiC enables the external load to be transferred from the Al matrix to SiC particles with higher strength, thus improving the fracture strength of Al–SiC composites. The hardness of SiC is significantly higher than that of the aluminum matrix, and this difference in hardness leads to the fracturing of Al grains along the bonding interface under greater stress, resulting in fine-grain strengthening of the aluminum matrix and improved tensile properties [20]. The Hall-Petch relationship can be used to calculate the increase in tensile strength resulting from grain refinement [21]:(2)$$\Delta \sigma_{GS}=k\left(d_{c}^{{-}_{2}^{1}}-d_{m}^{{-}_{2}^{1}}\right)$$
where $\Delta \sigma_{GS}$ (MPa) is the grain refinement strength increment, $d_{c}$ and $d_{m}$ (mm) are divided into the average grain size of the Al–SiC composite and Al matrix materials and *k* is a constant with a value of 0.1 MPa·mm^1/2^.

As shown in Figure 9, the addition of SiC particles can significantly enhance the strength of Ti/Al/Ti clad plates. With increasing SiC particle size, the tensile strength of the clad plate increases first and then decreases, and the elongation increases gradually. This is because the agglomeration of nanoscale particles becomes the defect source of the Al matrix, resulting in poor mechanical properties of the Al matrix. For large SiC particles, the strength of the aluminum matrix of HS_3_ is slightly lower than that of HS_1_ because under the condition of uniform particle distribution, the smaller the SiC particles are, the more obvious the strengthening effect on the aluminum matrix. On the other hand, the addition of SiC particles reduces the elongation. This is because fine particles can be fine-grain strengthened to prevent crack formation and dislocation movement, which reduces the co-deformation ability between the matrix and SiC. The cracks in the intermetallic compound layer and the number of aluminum-based aggregates are the highest in HS_0.5_ (Figure 5), and microcracks become the crack propagation source and preferred orientation during plastic deformation, so the elongation is the lowest. With increasing SiC particle size, the distribution of micro-SiC particles is more uniform, the elongation gradually increases, the microcracks in the HS_3_ alloy layer are reduced, the crack sources are reduced in number and large SiC particles can hinder crack propagation, so increasing the SiC particle size can effectively improve the elongation of Ti/Al–SiC/Ti clad plates.

### 3.5. Morphology of the Bonding Interface after Tensile Testing

Figure 10 shows the tensile fracture morphology of the Ti/Al–SiC/Ti composites. In Figure 10c,d, it can be clearly observed that the Ti/Al–SiC/Ti composite plate strengthened with 1 μm SiC has an obvious necking phenomenon before fracture, indicating that it has undergone a concentrated plastic deformation stage after uniform plastic deformation and shows obvious plastic fracture characteristics. Figure 10d shows the presence of many clustered dimples of various shapes and sizes in the Al layer. The results show that the many clustered dimples indicate that the Al matrix has a large plastic deformation and bears a high plastic strain. The measurement of plasticity in a material can be inferred from the characteristics of the ligament socket, including its shape, size and depth. It is observed that Figure 10d,f exhibits identical tensile fracture morphology. Furthermore, the presence of randomly distributed large and deep ligament sockets on the substrate surface indicates that the Ti/Al–SiC/Ti composite plate achieved complete metallurgical combination and mechanical bonding. Consequently, HS_1_ and HS_3_ demonstrate comparable tensile strength and elongation, as depicted in Figure 9. Based on the analysis of the HS_0.5_ fracture morphology as depicted in Figure 10b, it is evident that the absence of a discernible dimple structure is notable. Instead, the presence of numerous small cavities resulting from the diffusion of various elements and tiny dimples is observed. This observation suggests that mechanical bonding prevails, while the presence of adequate metallurgical bonding is lacking. Consequently, the material exhibits subpar mechanical properties. Additionally, Al has more slip than Ti; good deformation coordination makes it less likely to cause stress concentration and microcrack formation in the plastic denaturation process.

The toughness of the Ti/Al matrix and TiAl_3_ intermetallic compound layer is very different, and the intermetallic compound layer with poor toughness fractures first. The interfacial adhesion strength decreased, and the matrix could not continue to coordinate deformation. TiAl_3_ is a brittle phase that is prone to cracking during preparation. After annealing, the layer of intermetallic compound corresponding to HS_0.5_ is thicker, and there are more cracks and defects. Microcracks become the source of crack propagation in the tensile test, which leads to a decrease in the tensile properties of the composite and the phenomenon of cracking and delamination.

The HS_3_ sample has high plasticity, and the tear surface structure is “honeycomb” (Figure 10f). The intermetallic compounds showed obvious fracturing and cracking, and the bonding strength with the Al matrix decreased significantly. This is mainly due to the large SiC particles, which hinder the mutual diffusion of Ti–Al atoms at the interface, resulting in a thin intermetallic compound layer (3.8 μm), and the interface bonding strength of the Ti–Al matrix is reduced to 16.8 N/mm^−1^.

As seen from the tensile fracture morphology of the aluminum matrix of the Ti/Al–SiC/Ti composite plate (Figure 10b,d,f), the partial fracture morphology of HS_0.5_ shows brittle fracturing and agglomeration. The phenomenon of agglomeration can be effectively mitigated by increasing the size of SiC particles. This is because the agglomeration zone of SiC exhibits limited plasticity, which in turn reduces the co-deformation ability of Ti/Al under axial loading. Consequently, this leads to diminished tensile characteristics, such as in HS_0.5_. Therefore, during the tensile process, fractures preferably form in this region, and due to the high deformation capacity of the aluminum matrix, this tensile fracture pattern occurs and the matrix exhibits excellent tensile properties (Figure 9a). HS_1_ and the fracture morphology of the Al matrix of HS_3_’s increasing litter size increased gradually, and the toughening nest depth and quantity increased. The uniform dispersion of silicon carbide particles was the main cause of this difference.

The base materials of HS_1_ and HS_3_ exhibit identical plastic characteristics, resulting in comparable fracture morphologies characterized by equal dimensions, depths and a homogeneous distribution of the dimple structure. The metallurgical bonding ability and tensile fracture resistance ability at the Ti/Al–SiC interface during the fabrication of a Ti/Al–SiC/Ti composite plate by the powder-in-tube method can be determined by analyzing the thickness and shape of the intermetallic compound layer. The intermetallic compound layer, as depicted in Figure 5e, exhibits uniformity, completeness and continuity, hence enhancing its capacity for interface bonding and resistance against tensile fracture. Increased presence of microcracks and inadequate metallurgical bonding within the composite material (Figure 5c,g) render it more susceptible to failure under applied loads. Compared with HA500 (Figure 5a), the plasticity of the Al matrix increases, but the fracture of the intermetallic compound layer at the bonding interface decreases the overall plasticity of the clad plate.

## 4. Conclusions

In this paper, a novel preparation method of Ti/Al composite plate is presented—Powder-in-tube method (PIT method). Ti/Al-SiC/Ti composite plates were successfully prepared by designing core materials. The effects of SiC concentration and particle size on interfacial diffusion, tensile and stripping properties were studied, the following conclusions were drawn.

(1)Ti/Al–SiC/Ti clad plates were prepared by the powder-in-tube method, which can significantly improve the mechanical properties of clad plates and simplify the production process of clad plates;(2)Compared with the HA500 sample, the addition of 6 wt.% SiC particles enhanced the diffusion and thickened the intermetallic compound layer. HS_1_ corresponds to the best intermetallic compound layer thickness; the cracks are reduced compared to HS_0.5_, the diffusion layer thickness is greater than HS_3_ and no cracks or defects appear. The aggregation of nanoscale SiC particles has no significant effect on the obstruction of element diffusion, and the intermetallic compounds in the HS_0.5_ sample are thicker and have more cracks. The large SiC particle size in HS_3_ inhibits the diffusion of Ti and Al elements, and the thickness of the intermetallic compound layer is small;(3)The XRD results show that the diffusion layer is mainly a brittle TiAl_3_ phase. In the peel strength test, the TiAl_3_ content on the Ti side decreases with increasing grain size, while the TiAl_3_ content on the Al side has no significant difference with increasing grain size. After adding SiC particles to Ti/Al/Ti composite plates, the interface peeling strength of composite plates is 24.1 N/mm (HS_0.5_), 31.5 N/mm (HS_1_) and 16.7 N/mm (HS_3_), respectively, and all have different degrees of decrease;(4)Tensile cracking originates from the SiC agglomeration zone’s poor plasticity. SiC size and dispersion must be controlled to ensure Ti/Al–SiC/Ti composite plate contact bonding. The SiC agglomeration zone prevents dislocation migration during plastic deformation, increasing tensile strength. Large SiC particle agglomerates fracture during plastic deformation and lower the tensile strength of the Ti/Al–SiC/Ti composite plate. Thus, as SiC particle diameter rises, the tensile strength of the Ti/Al–SiC/Ti composite plate first increases and then drops from 270 MPa (HS_0.5_) to 305 MPa (HS_1_) to 296 MPa (HS_3_).

## Figures and Tables

**Figure 1 materials-16-05986-f001:**
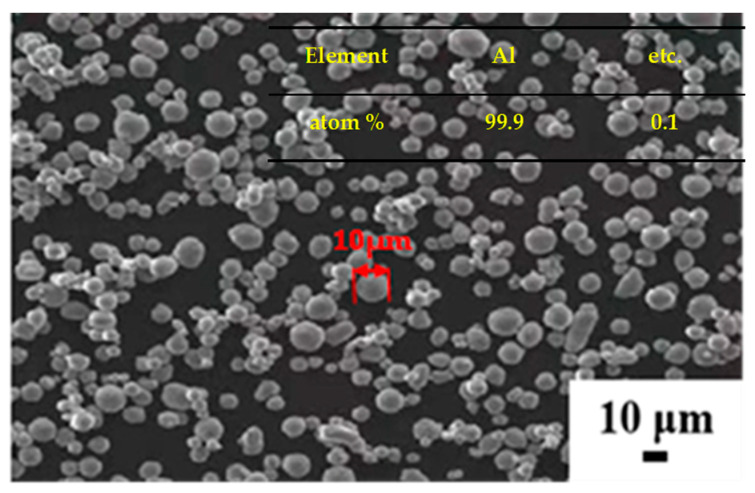
SEM image of atomized Al particles.

**Figure 2 materials-16-05986-f002:**
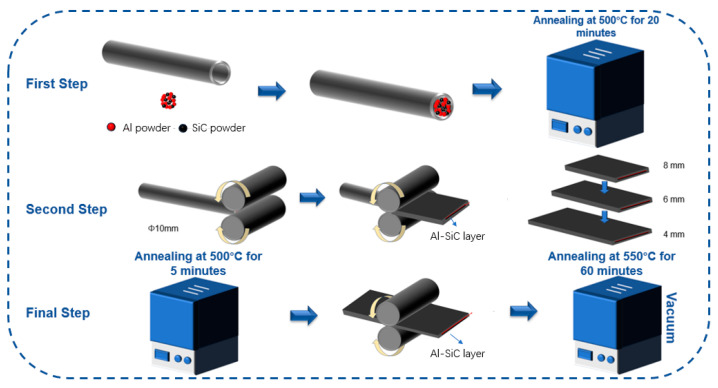
Schematic process of Ti/Al–SiC/Ti clad plate.

**Figure 3 materials-16-05986-f003:**
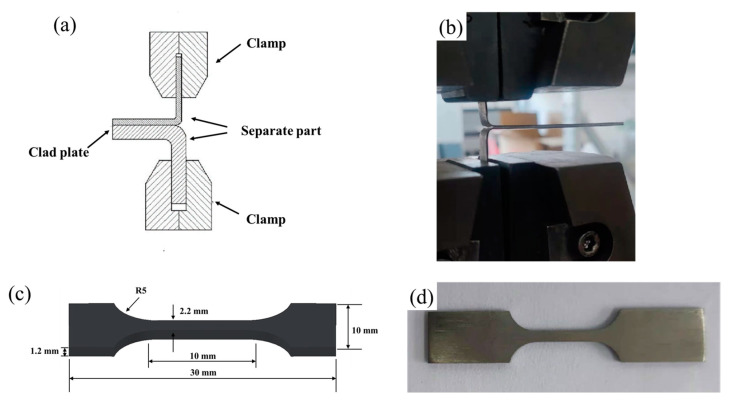
Tensile and interface stripping experiment of Ti/Al–SiC/Ti composite plate: (**a**) diagram of peeling test; (**b**) real image from peeling test; (**c**) diagram of tensile sample; (**d**) real image from tensile sample.

**Figure 4 materials-16-05986-f004:**
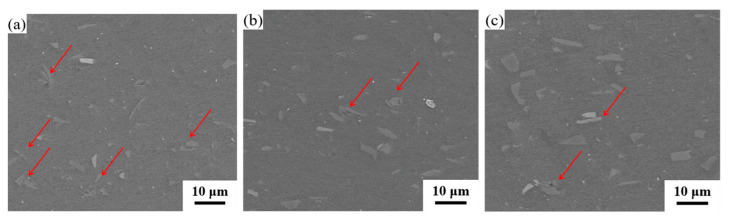
SEM image of Al–SiC Composites with Different Sizes: (**a**) HS_0.5_; (**b**) HS_1_; (**c**) HS_3_.

**Figure 5 materials-16-05986-f005:**
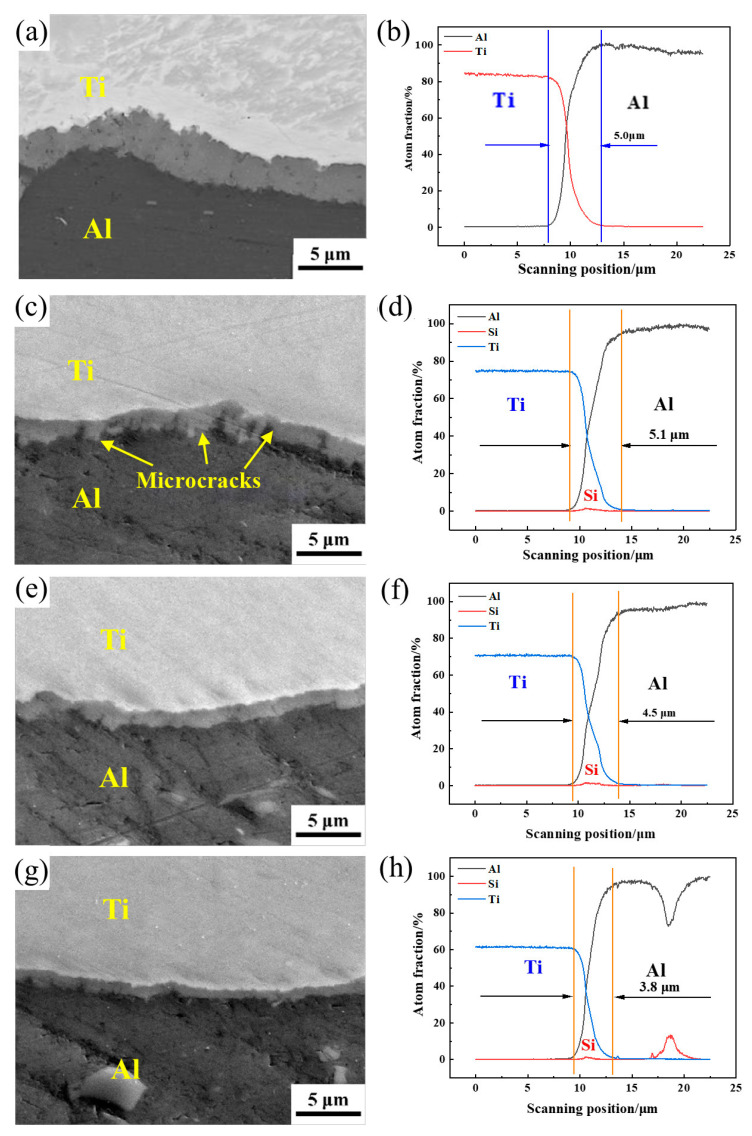
EPMA morphology of Al–SiC Composites with Different Sizes: Interface Morphology and Element Distribution of Ti/Al–SiC/Ti clad plates: (**a**) HA500; (**c**) HS0.5; (**e**) HS1; (**g**) HS3; (**b**) Line chart of HA500; (**d**) Line chart of HS0.5; (**f**) Line chart of HS_1_; (**h**) Line chart of HS_3_.

**Figure 6 materials-16-05986-f006:**
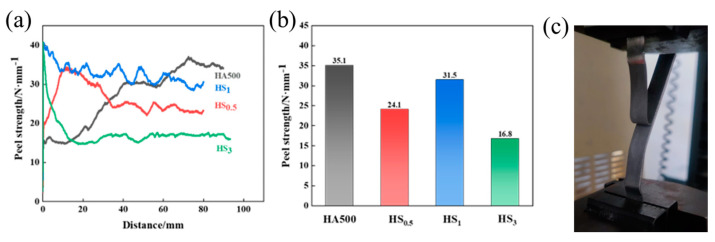
The peeling strength of the Ti/Al–SiC/Ti clad plate: (**a**) peeling curve of Ti/Al–SiC/Ti clad plates with various SiC diameters; (**b**) average peeling strength of Ti/Al–SiC/Ti clad plates with various SiC diameters; (**c**) the real image of peeling fracture.

**Figure 7 materials-16-05986-f007:**
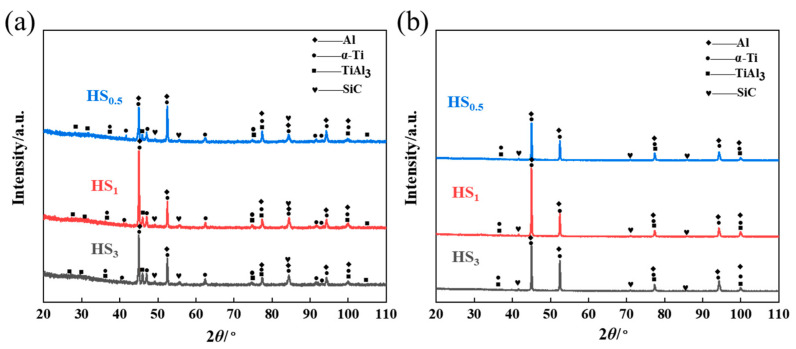
XRD patterns of the peeling surface of the Ti/Al–SiC/Ti clad plate sample: (**a**) Ti side; (**b**) Al side.

**Figure 8 materials-16-05986-f008:**
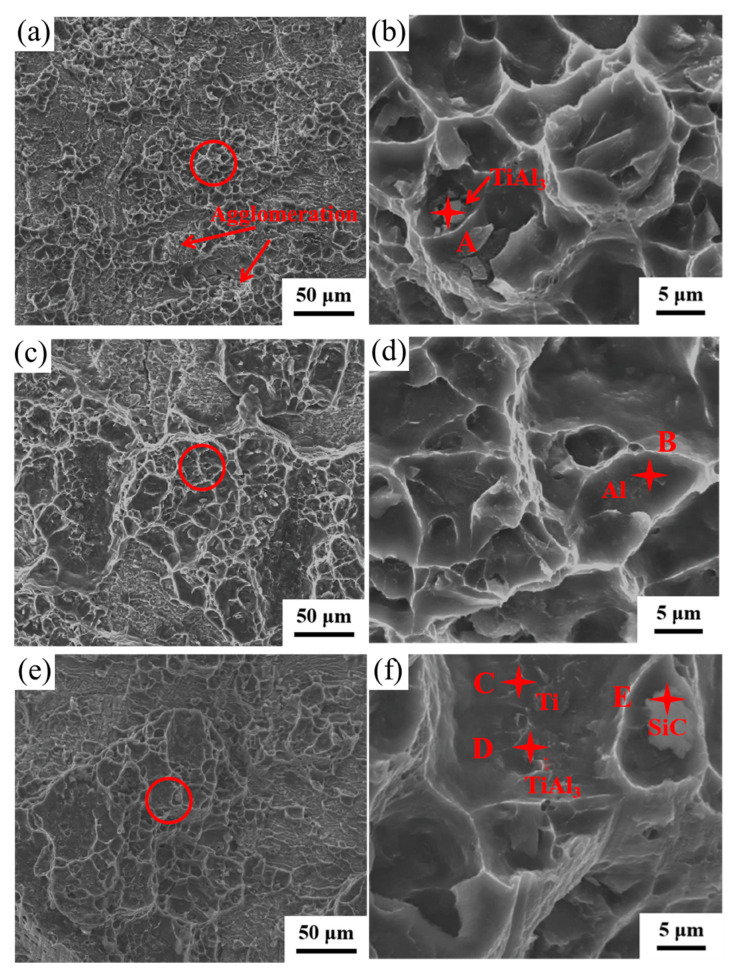
Peeling morphology of Ti/Al–SiC/Ti clad plates: (**a**) HA_0.5_, (**b**) HA_1_, (**c**) HA_3_, (**d**) Enlarged view of the circle in (**a**), (**e**) Enlarged view of the circle in (**b**), and (**f**) Enlarged view of the circle in (**a**).

**Figure 9 materials-16-05986-f009:**
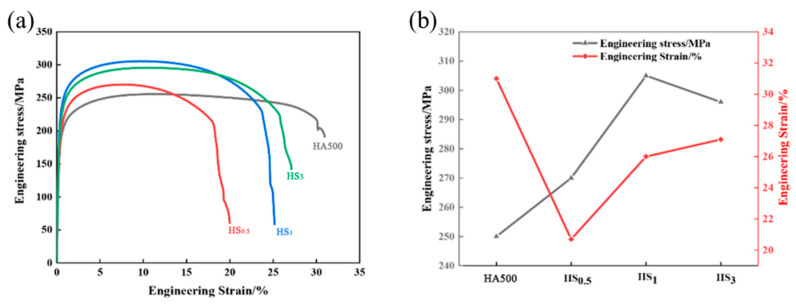
Mechanical properties of the Ti/Al–SiC/Ti clad plates: (**a**) Tensile curve of Ti/Al–SiC/Ti clad plates; (**b**) Engineering stress and strain of Ti/Al–SiC/Ti clad plates.

**Figure 10 materials-16-05986-f010:**
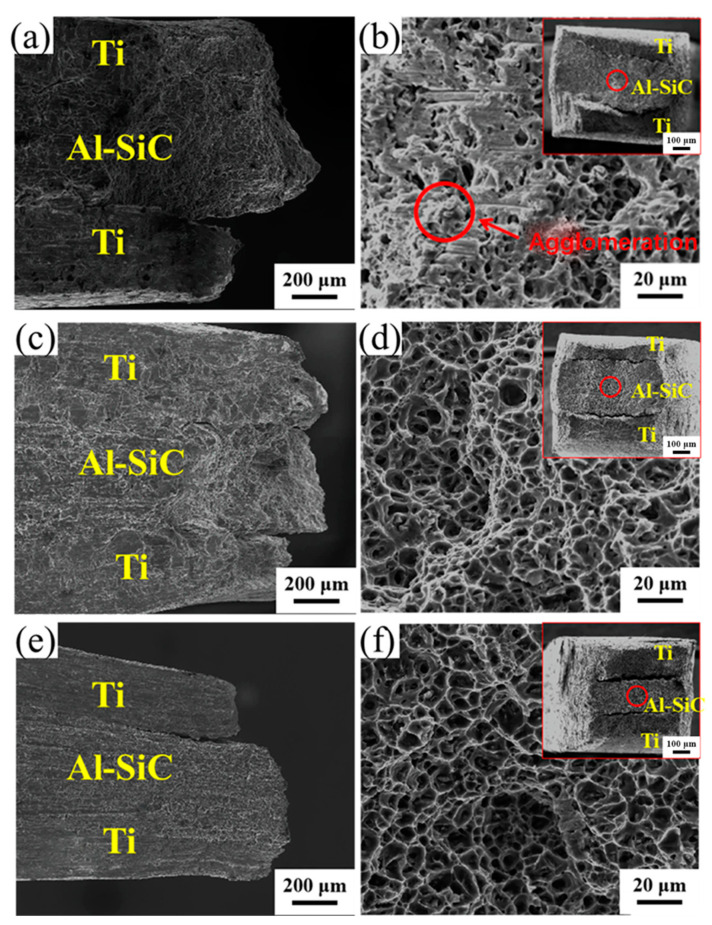
Tensile fracture morphology of Ti/Al–SiC/Ti clad plates: (**a**) HS_0.5_; (**b**) Al matrix of HS_0.5_; (**c**) HS_1_; (**d**) Al matrix of HS_1_; (**e**) HS_3_; (**f**) Al matrix of HS_3_.

**Table 1 materials-16-05986-t001:** The XRD quantitative results. of HA_0.5_, HA_1_ and HA_3_. (mass %).

Side	Sample	α-Ti	Al	TiAl_3_	SiC
Ti	HS_0.5_	33.30	19.70	8.40	38.60
	HS_1_	26.50	24.20	9.70	39.60
	HS_3_	42.90	10.50	7.80	38.80
Al	HS_0.5_	55.30	5.10	8.40	31.10
	HS_1_	42.60	11.50	7.70	38.30
	HS_3_	46.40	6.70	13.00	33.90

**Table 2 materials-16-05986-t002:** EDS microanalysis of HA_0.5_, HA_1_ and HA_3_.

Portion	Ti (Atom %)	Al (Atom %)	Si (Atom %)	C (Atom %)
A	67.92	27.08	-	-
B	2.54	97.46	-	-
C	91.22	7.99	-	-
D	72.72	27.28	-	-
E	-	-	51.38	48.62

**Table 3 materials-16-05986-t003:** Tensile properties of the Ti/Al–SiC/Ti clad plates.

Samples	Tensile Strength/MPa	Yield Strength/MPa	Elongation/%
HA500	250	182	31
HS_0.5_	270	195	20.7
HS_1_	305	220	26
HS_3_	296	211	27.1

## Data Availability

Not applicable.

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
