# Peer review of "Microstructure and Mechanical Properties of Ti/Al–SiC/Ti Clad Plates Prepared via the Powder-in-Tube Method"

_materials, 2023, doi:10.3390/ma16175986_

Round 1

Reviewer 1 Report

Dear Authors,

Your I enjoyed reading your research, several issues need to be adressed:

- in the title you state that the microstructure is investigated, but no clear micrographs depicting the microstructure are presented. One could relate to fig. 3 regarding the microstructure, but its quality is rather poor. Enhancing the constrast is required. I would like to see the effective microstructure, in a cross-section, for the PIT specimens or the cladded samples, similar to those presented in the fracture morphology of the tensile specimens.

- there are several issues in English style and grammar that I have marked in an attached document. Please adress them. The teminology used is inappropriate. Also several sentences are repeated, and the dimensions for the titanium pipe are wrong, the wall thickness is once 1mm and then 10mm. I think the outer diameter is 10mm and wall thickness is 1mm. Please check.

- I strongly advise to rewrite the materials and methods section, more precise the methods used. They are poorly described and insufficient information presented.

- the TA500 sample coding was not presented in the materials and methods

- line 99, scanning morphology -> particle morphology observed by SEM

line 102, expose new metals -> unclear, perhaps expose the bare metals?

line 110, no edge fracture -> unclear, revise.

fig. 3 -> needs contrast enhancement

line 181, the elastic modulus of SiC is 475MPa? Please check, improbable. This compound has very high hardness and elastic modulus.

fig. 4 scanning morphology - revise

line 198, how does a ductile fracture becomes brittle? the material behavior is ductile or brittle, the fracture is a consequence of material behavior. please revise.

line 201, the crack propagation path of the intermetallic compound layer is short - revise, 

line 227-228, revise the sentence regarding the XRD

the XRD results needs further discussion, aside phase identification.

line 341, crack diffusin path, revise

figure 9, tensile morpholoy - revise

check references to fig. 8 starting from line 392. It must be fig. 9.

- the discussion starting from line 407 in terms of plasticity must be revised. The plasticity of the matrix is the same, the load distribution is different and failure occurs accordingly. Direct your disussion towards this aspect.

- line 419, crack density - revise

Your research is interesting and I support its publication when the issues are adressed.

My best regards.

Please see the highlighted regions in the attached document.

Reviewer 2 Report

Review report: Microstructure and mechanical properties of Ti/Al-SiC/Ti clad 2 plates prepared via the powder-in-tube method. Work is presented well with good publishing quality and can be accepted after the following corrections:  

1.       Abstract: Add some quantitative results related to mechanical testing at end of the abstract section.

2.       Introduction: In place of citing multiple references, explain the individual work of the author and try to make a bridge between current and previous work. Refer to some recently published work: https://doi.org/10.1007/s12666-016-0977-6; https://doi.org/10.1007/s12633-017-9710-2

3.       Novelty and application: Add a separate section for novelty and application of work.

4.       Materials and methods: Section is presented well but need some corrections. Add a detail of experimental set up instead of a schematic image. Also add the parameters and provide the mechanical properties of the used material.

5.       In SEM image add the particle size measurement and EDS.

6.       In element diffusion, the EDS is required. Also add the particle size distribution.

7.       In Fig. 4, mark the line scan in main image. Also improve the quality of SEM image.

8.       How was the peeling strength calculated? Also variation in peeling strength is not clear from the particular discussion.

9.       In XRD add the quantitative results for each phase.

10.    Agglomeration is not clear from the image. This is a single particle. Add eth image of the fractured sample.

11.    Similarly, in tensile strength add the sample image after fracture. Also, mention the standard used.

12.    Try to relate the tensile properties with microstructure and a separate section of discussion.

13.    Add discussion related to dimples and voids on fracture surfaces.

14.    Shorten the length of the conclusion section. 

NA

Reviewer 3 Report

The article presents the results of a study of Ti/Al-SiC/Ti clad plates obtained by the powder-in-tube method. The article contains original scientific results and will be of interest to the reader. There are comments:

1. Fig. 3, 6 and 8 need to be increased. As presented, they are not clear.

2. Fig. 4 shows the concentration distribution curves of Ti, Al and Si at the interface. The diffusion zone values are given with an accuracy of 0.001 µm. Over the entire length of the composite, the size of this zone will not be sized with such accuracy. I recommend that this size be averaged and given with an accuracy of 0.1 microns.

3. It is necessary to indicate the dimensions of the samples that were tested for tear strength and tensile strength. It's even better to show them a photo.

4. In fig. 7 shows the structural components (Ti, Al, TiAl3, SiC). To establish their location, it is additionally necessary to provide the results of EDA at the indicated points. In the presented form, there is no evidence of their location.

5. Lines 336-343. It is not appropriate to talk about the number of cracks, because the article does not provide evidence. Similar remark on Conclusion 2. Please provide evidence.

6. Conclusion 3. Taking into account that the diffusion layer is no more than 5 μm, in addition to the XRD results, it is necessary to present the EDA results.

Round 2

Reviewer 3 Report

The authors answered all questions and made significant additions. The paper has gotten a lot better. I recommend publishing.